# Perceptions, behaviours and barriers towards exercise practices in inflammatory bowel disease

**Jonathan Sinclair[1], Johanne Brooks-Warburton[2], Lindsay Bottoms [3]\***

**1** Research Centre for Applied Sport, Physical Activity and Performance, School of Sport & Health Sciences, Faculty of Allied Health and Wellbeing, University of Central Lancashire, Lancashire, United Kingdom, **2** School of Life and Medical Sciences, University of Hertfordshire, Hertfordshire, United Kingdom, **3** Centre for Research in Psychology and Sport Sciences, School of Life and Medical Sciences, University of Hertfordshire, Hertfordshire, United Kingdom

\* l.bottoms@herts.ac.uk

**Data Availability Statement:** The data relevant to this study are available from UCLAN's data repository at DOI: 10.17030/uclan.data.00000434 (https://uclandata.uclan.ac.uk/434/).

## Abstract

Inflammatory bowel disease (IBD), a chronic disease affecting the digestive tract, has a significant impact on health-related quality of life. Pharmaceutical treatment is typically adopted, yet exercise is increasingly becoming recognized as an adjunct therapy. This study aimed to explore the perspectives, behaviours, and barriers of IBD patients in terms of their exercise habits. A 16-item closed-ended questionnaire was completed by 463 adult IBD patients (Ulcerative colitis = 57.02%, Crohn's dis-ease = 40.60% and Other = 2.38%) (Female = 76.67%, Male = 22.46 and Non-binary = 0.86%). The questionnaire was divided into three sections: baseline/demographic characteristics, disease characteristics, and exercise perceptions, beliefs, and behaviours. Significantly (P<0.001) more participants (63.07%) reported that they engage regularly with exercise compared to those who do not; however, engagement was significantly lower in female patients (59.72%) compared to males (74.04%). Respondents also rated significantly (P<0.001) that a combination of factors prevents engagement in exercise (74.30%). Moderate intensity exercise was the predominant (P<0.001) aerobic modality (39.04%), the majority (P<0.001) response was that patients undertake no resistance training (27.74%), and significantly more (P<0.001) patients indicated that they don't know whether resistance training can influence IBD either positively (57.53%) or negatively (62.33%). Whilst it is encouraging that IBD patients are engaging regularly with exercise, the reduced levels of engagement in females and lack of knowledge/ engagement with resistance training, indicate that future implementation and educational developments are necessary to enhance exercise in females and resistance training engagement in all IBD patients.

## Introduction

Inflammatory bowel diseases (IBD), including ulcerative colitis (UC) and Crohn's disease (CD), are chronic conditions affecting the digestive tract [1, 2]. UC targets the colon's lining,

**Funding:** The author(s) received no specific funding for this work.

**Competing interests:** The authors have declared that no competing interests exist.

while CD induces inflammation throughout the digestive tract [2]. Primary IBD symptoms include fatigue, pain, and diarrhea, potentially leading to malabsorption, low bone mineral density, and muscle loss [3–5]. Additional extraintestinal manifestations such as osteoarthritis, uveitis, iritis, episcleritis, erythema nodosum, and pyoderma gangrenosum may arise [3, 6]. Global incidence of IBD is rising [7], with the annual economic toll exceeding $31 billion in the United States alone [8]. IBD profoundly affects quality of life, encompassing psychological well-being, body image, and social relationships [9]. Additionally, it hampers labour force participation, exacerbating the disease's fiscal impact [10]. The clinical spectrum of IBD varies widely, spanning from asymptomatic quiescence to potentially life-threatening severe illness [1].

IBD clinical management aims for remission, improved quality of life, and complication prevention [11], often relying on pharmaceutical intervention [12]. Traditional options include Azathioprine, Mercaptopurine, Infliximab, Tofacitinib, Ustekinumab, Vedolizumab, Adalimumab, steroids, and oral 5-aminosalicylates [11, 13]. Despite pharmaceutical advancements, surgery remains commonplace for IBD treatment, with recent findings indicating an 8.8% surgery rate for CD and 7.5% for UC [14].

The multifactorial pathogenesis of IBD involves intricate interactions among genetics, environment, lifestyle, and the immune system [10]. Notably, developed nations exhibit the highest IBD prevalence, suggesting a connection between Westernized lifestyles and IBD etiology [15–17]. Sedentary Western lifestyles, marked by insufficient exercise [18], have been identified as a risk factor for IBD development [19]. Exercise is increasingly advocated as an adjunct therapy for IBD [20]. Intervention studies reveal the efficacy of low-moderate intensity aerobic exercise in enhancing IBD patients' quality of life, reducing inflammation, fatigue, and improving bone mineral density and psychological well-being, with minimal adverse effects [21–26]. While the precise biological mechanism remains unclear, alterations in gut microbiota and the impact of exercise on immunological and inflammatory processes are suggested pathways [27].

Current guidelines for exercise in adults with IBD recommend 20–60 minutes of low-moderate intensity exercise on 2–5 days per week [28]. However, these guidelines were established when research on exercise efficacy in IBD was in its early stages, relying on theoretical foundations from observations in healthy individuals [29]. Presently, IBD exercise recommendations overlook potential therapeutic benefits from contemporary modalities like high-intensity exercise [20, 30] and resistance training [31, 32].

Despite recognizing the positive impact of exercise in IBD, patients often exhibit reduced activity levels post-diagnosis [33]. Varying studies report different activity rates: in Italy, 20.6% [34] and 25.6% [35] engaged in regular physical activity. In Canada, Crohn's patients were 58.0%, 24.1%, and 17.9% inactive, moderately active, and active, while UC patients were 53.6%, 25.4%, and 21.0%, respectively [36]. The Crohn's and Colitis Foundation in the UK found that 66% of patients exercised, with 32% doing so daily, 57% weekly, 5% monthly, and 5% less than monthly [1]. In the US, 16.4% never exercised, 32.8% exercised 1–2 times weekly, 23.6% 3–4 times, and 18.0% exercised >4 times weekly [37]. Another study reported that 17% were highly active, 50% minimally active, and 33% inactive [29]. While existing analyses quantify activity, limited attention is given to IBD patient perceptions, behaviors, and barriers towards exercise. Further research is needed to understand these factors across patient subgroups and other potentially important disease/ demographic factors age, disease activity, gender, and diverse training modalities, aiding in tailored interventions for sustained remission and improved quality of life in IBD patients. Insights gained from such research can inform strategies promoting long-term well-being and provide a foundation for targeted interventions.

Pharmaceutical intervention is the primary IBD management approach [12], yet these therapies often require continuous use [27], come with significant side effects [38], and exhibit low adherence rates [39]. Seeking alternative treatments is increasingly common [40], with up to 56% of patients exploring complementary modalities [41]. Prescribing exercise as an adjunct therapy could enhance health-related quality of life for IBD patients at a lower economic cost. A comprehensive understanding of exercise perceptions, beliefs, and barriers is crucial for implementing new exercise therapies. This investigation aims to provide insights into IBD patients' perceptions, behaviours, and barriers related to exercise, contributing valuable information for the development and integration of exercise-based interventions in this population.

## Materials and methods

### Ethical considerations

The study was approved by the Institutional ethical board of the University of Central Lancashire reference: HEALTH 0423. All participants provided informed consent in accordance with the Declaration of Helsinki.

### Study design and participants

This study used an observational and quantitative approach to achieve its objectives. IBD patients anonymously completed a self-administered online questionnaire hosted on the Jisc platform (Jisc, UK). Recruitment was undertaken in the in the United Kingdom through Twitter and Facebook based social media platforms, using general and research specific accounts belonging to Crohn's and Colitis UK. Participants were eligible to participate in the study if they were over 18 years of age and had been diagnosed with Crohn's disease (CD), ulcerative colitis (UC), or another IBD condition. The survey was available online for 3 months, from March 1st 2023 to June 1st 2023. A total of 463 responses were received (UC = 264, CD = 188, Other = 11) after the survey period ended.

### Questionnaire

The questionnaire was created after reviewing scientific studies on exercise in IBD patients. This facilitated the identification of relevant topics and development of questions and response options. In accordance with our previous questionnaire-based analyses, four research and clinical experts in the field of IBD reviewed the survey and provided feedback to improve its face validity [40, 42]. Two of the aforementioned experts and one of the authors of the paper importantly have lived experience of IBD. This feedback was incorporated into the questionnaire design before it was submitted for ethical approval. The options for respondents in relation to medication questions were developed based on the current guidelines of Chron's & Colitis UK and the British Society of Gastroenterology consensus guidelines, in consultation with the aforementioned clinical professionals from within this field [43].

The questionnaire, excluding the information page and consent options, featured 16 close-ended questions divided into three sub-sections. Section 1 related to participants' baseline demographics, such as gender, age, smoking, and alcohol use. Section 2 explored disease characteristics, such as the specific IBD condition, disease activity, and current medication. The specific IBD condition featured three options, Ulcerative Colitis, Crohn's and Other as although UC and CD represent the overwhelming majority of cases of IBD, there are other inflammatory bowel conditions, and it is also possible for patients who suffer from both UC and CD. Section 3 considered exercise perceptions, beliefs, and behaviours.

The section relating to exercise perceptions, beliefs, and behaviours included a series of questions arranged as following in chronological order; whether patients exercise regularly, the specific type of aerobic exercise they engaged in, their perceptions regarding potential positive or negative impact of aerobic exercise on IBD, the type of resistance training they performed, and their perceptions regarding potential effects of resistance training on IBD. Additionally, the questionnaire asked about the avoidance of certain types of exercise. Patients who reported that they do not engage in regular exercise regularly were directed to the question about avoiding certain kinds of exercise. The questionnaire is presented in Supplementary materials (S1–S3 Appendices).

## Statistical analyses

The data from the questionnaire were inputted into SPSS v27 (IBM), and categorical data for each survey question were coded. When reporting proportions for categorical variables, both the total count (N) and the corresponding percentages (%) were provided. In questions which allowed respondents to select multiple options i.e. questions 7, 15 and 16 (as shown in S3 Appendix), for those who did so, a single 'combination' category was created. One-sample chi-square ($X^2$) goodness of fit tests were employed for all questions in the exercise perceptions, beliefs, and behaviours section, as well as the questions related to smoking behaviour, alcohol consumption, and current medication [40]. These tests were used to compare the proportions of participants who selected each response option. In addition, two-way Pearson chi-square tests of independence were used to perform bivariate comparisons. Specifically, these tests were conducted to assess differences in responses to each question based on baseline/demographic variables, gender, age, specific IBD condition and disease activity [40]. For the sake of brevity, only the two-way Pearson chi-square tests of independence that yielded statistically significant results were presented in the results. Statistical significance for all analyses was set at the $P<0.05$ level.

## Results

### Baseline/demographic characteristics

The baseline/demographic characteristics of the study population are described in Table 1. Overall, there were 463 eligible patients in the survey, of whom 76.67% were female, 22.46% male and 0.86% non-binary. In addition, 57.02% of patients had UC, 40.60% CD and 2.38% another IBD condition. Overall, 30.45% of patients were in remission, 36.50% experienced mild, 27.86%, moderate and 5.18% severe symptoms.

### Disease characteristics

The disease characteristics of the study population are described in Table 2. One-sample chi-square goodness of fit tests showed that there were significant differences in responses for 'Medication' ($X^2_{(11)}$ = 440.16, $P<0.001$) with significantly more participants taking a combination of medications. Finally, there were also significant differences in responses for both 'Smoking behaviour' ($X^2_{(1)}$ = 316.82, $P<0.001$) with the majority of participants being non-smokers and also 'Alcohol' consumption ($X^2_{(6)}$ = 955.28, $P<0.001$) which showed that the majority of participants consumed no alcohol. No further statistically significant findings were observed.

**Table 1. Baseline/demographic characteristics.**

| Variables | N | % |
|---|---|---|
| *Gender* | | |
| Female | 355 | 76.67 |
| Male | 104 | 22.46 |
| Non-binary | 4 | 0.86 |
| *Age* | | |
| 18–25 | 45 | 9.72 |
| 26–30 | 44 | 9.50 |
| 31–35 | 59 | 12.74 |
| 36–40 | 84 | 18.14 |
| 41–45 | 65 | 14.04 |
| 46–50 | 66 | 14.25 |
| 51–55 | 45 | 9.72 |
| 56–60 | 24 | 5.18 |
| 61–65 | 22 | 4.75 |
| 65+ | 9 | 1.94 |
| *Disease* | | |
| UC | 264 | 57.02 |
| CD | 188 | 40.60 |
| Other | 11 | 2.38 |
| *Disease activity* | | |
| Remission | 141 | 30.45 |
| Mild | 169 | 36.50 |
| Moderate | 129 | 27.86 |
| Severe | 24 | 5.18 |

## Exercise perceptions, beliefs and behaviours (one-sample analyses)

The exercise perceptions, beliefs and barriers of the study population are described in Tables 3 and 4. There were significant differences in responses for 'Do you exercise regularly' ($X^2_{(1)}$ = 31.62, P<0.001) which showed that significantly more participants are regularly engaged with exercise activities and also 'What would/ does prevent you from engaging in exercise' ($X^2_{(7)}$ = 1673.17, P<0.001) which showed that significantly more participants indicated that a combination of factors prevent engagement in exercise. Of those who exercise regularly there was a significant difference in responses for 'Which type of aerobic exercise do you mostly do' ($X^2_{(3)}$ = 84.66, P<0.001) which revealed that significantly more participants engaged in moderate intensity exercise.

In addition, there were significant differences in responses for 'Do you believe aerobic exercise can influence Inflammatory bowel disease in a positive way' ($X^2_{(2)}$ = 126.30, P<0.001) which showed that significantly more participants felt that aerobic exercise has a positive influence on Inflammatory bowel disease and for 'Do you believe aerobic exercise can influence Inflammatory bowel disease in a negative way' ($X^2_{(2)}$ = 7.24, P = 0.027) which revealed that significantly more participants indicated that they do not know if aerobic exercise can negatively influence Inflammatory bowel disease.

There was a significant difference in responses for 'Which type of resistance exercise do you mostly do' ($X^2_{(5)}$ = 68.49, P<0.001) which revealed that significantly more participants engaged in no resistance training. In addition, there were significant differences in responses

**Table 2. Disease characteristics (\* = significant chi-squared test).**

| Variables | N | % | |
|---|---|---|---|
| *Medication* | | | \* |
| None | 59 | 12.74 | |
| Adalimumab | 36 | 7.78 | |
| Azathioprine and Mercaptopurine | 35 | 7.56 | |
| Biologic Drugs | 5 | 1.08 | |
| Infliximab | 13 | 2.81 | |
| Steroids | 39 | 8.42 | |
| Tofacitinib | 1 | 0.22 | |
| Ustekinumab | 26 | 5.62 | |
| Vedolizumab | 19 | 4.10 | |
| Aminosalicylates (5-ASAs) | 26 | 5.62 | |
| Other | 54 | 11.66 | |
| Combination | 150 | 32.4 | |
| *Smoke* | | | \* |
| Yes | 40 | 8.64 | |
| No | 423 | 91.36 | |
| *Alcohol (units/week)* | | | |
| None | 281 | 60.69 | |
| 1–2 | 118 | 25.49 | |
| 3–4 | 20 | 4.32 | |
| 5–6 | 15 | 3.24 | \* |
| 7–9 | 9 | 1.94 | |
| 10+ | 7 | 1.51 | |
| Other | 13 | 2.81 | |

for 'Do you believe resistance exercise can influence Inflammatory bowel disease in a positive way' ($X^2_{(2)}$ = 124.30, P<0.001) and 'Do you believe resistance exercise can influence Inflammatory bowel disease in a negative way' ($X^2_{(2)}$ = 127.75, P<0.001) which revealed that significantly more participants indicated that they do not know if resistance exercise can positively or negatively influence Inflammatory bowel disease.

There was a significant difference in responses for 'Do you avoid certain types of exercise' ($X^2_{(9)}$ = 633.21, P<0.001), which revealed that significantly more participants did not avoid any exercise modality. No further statistically significant findings were observed.

## Exercise perceptions, beliefs and behaviours (two-way cross-tabulation analyses)

The exercise perceptions, beliefs and behaviours of the study population as a function of the baseline/demographic groups that exhibited statistical significance when examined using two-way cross-tabulation analyses are described in S1A–S1E Appendix. Those that did not exhibit statistical significance are presented collectively in S2 Appendix. These tables are presented as appendices owing to their size.

A significant difference ($X^2_{(2)}$ = 8.17, P = 0.004) in the responses to 'Do you exercise regularly' as a function of 'Gender' was observed, with a greater proportion of males indicating that they engage with exercise on a regular basis. There was also a significant difference ($X^2_{(10)}$ = 26.72, P = 0.003) in the responses to 'Which type of resistance exercise to you mostly do' as a

**Table 3. Exercise perceptions, behaviours and barriers (\* = significant chi-squared test).**

| Variables | N | % | |
|---|---|---|---|
| *Do you exercise regularly?* | | | \* |
| Yes | 292 | 63.07 | |
| No | 171 | 36.93 | |
| *What would/ does prevent you from engaging in exercise?* | | | \* |
| Combination of factors | 344 | 74.30 | |
| Fatigue | 64 | 13.82 | |
| Fear of increased toilet urgency | 28 | 6.05 | |
| Other | 20 | 4.32 | |
| Pain during exercise | 3 | 0.65 | |
| Fear of triggering a flare up | 2 | 0.43 | |
| Lack of scientific evidence | 1 | 0.22 | |
| Fear of increased abdominal pain | 1 | 0.22 | |
| *Which type of aerobic exercise to you mostly do?* | | | \* |
| Moderate intensity | 114 | 39.04 | |
| Low intensity | 96 | 32.88 | |
| Vigorous | 72 | 24.66 | |
| None | 10 | 3.42 | |
| *Do you believe aerobic exercise can influence Inflammatory bowel disease in a positive way?* | | | \* |
| Yes | 168 | 57.53 | |
| Don't know | 111 | 38.01 | |
| No | 13 | 4.45 | |
| *Do you believe aerobic exercise can influence Inflammatory bowel disease in a negative way?* | | | \* |
| Don't know | 119 | 40.75 | |
| Yes | 87 | 29.79 | |
| No | 86 | 29.45 | |
| *Which type of resistance exercise to you mostly do?* | | | \* |
| None | 81 | 27.74 | |
| Bodyweight | 73 | 25.00 | |
| Free weights | 59 | 20.21 | |
| Machines | 34 | 11.64 | |
| Other | 24 | 8.22 | |
| Structured classes | 21 | 7.19 | |

function of 'Gender' was observed, with a greater proportion of males utilizing machine-based resistance training and correspondingly a greater proportion of females attending structured resistance training classes.

There was also a significant difference ($X^2_{(9)}$ = 25.13, P = 0.003) in the responses to 'Which type of aerobic exercise to you mostly do' as a function of 'Disease activity', with a greater proportion of those with severe symptoms engaging in no aerobic exercise. There was also a significant difference ($X^2_{(9)}$ = 34.52, P = 0.032) in the responses to 'What would/ does prevent you from engaging in exercise' as a function of 'Disease activity', with a greater proportion of those with severe symptoms indicating that fear of increased toilet urgency prevents engagement in exercise and those in remission denoting that other factors were responsible. No further statistically significant findings were observed.

**Table 4. Exercise perceptions, behaviours and barriers (* = significant chi-squared test).**

| Variables | N | % | |
|---|---|---|---|
| ***Do you believe resistance exercise can influence Inflammatory bowel disease in a positive way?*** | | | * |
| Don't know | 168 | 57.53 | |
| Yes | 110 | 37.67 | |
| No | 14 | 4.79 | |
| ***Do you believe resistance exercise can influence Inflammatory bowel disease in a negative way?*** | | | * |
| Don't know | 182 | 62.33 | |
| No | 84 | 28.77 | |
| Yes | 26 | 8.90 | |
| ***Do you avoid certain types of exercise?*** | | | * |
| No | 132 | 45.21 | |
| Combination | 91 | 31.16 | |
| Vigorous intensity | 31 | 10.62 | |
| Other | 16 | 5.48 | |
| Structured weights classes | 6 | 2.05 | |
| Free weights | 5 | 1.71 | |
| Low intensity | 4 | 1.37 | |
| Machine based weightlifting | 3 | 1.03 | |
| Moderate intensity | 2 | 0.68 | |

## Discussion

The aim of the current investigation was to explore the perceptions, behaviours and barriers of IBD patients in relation to their exercise practices. To the authors knowledge this represents the first exploration of IBD patients' exercise perceptions, behaviours and barriers in different patient subgroups and other potentially important disease/ demographic factors. This investigation therefore provides information that may be important for the implementation of future tailored initiatives seeking to improve disease management and quality of life.

Importantly, the current investigation showed that significantly more participants are regularly engaged with exercise activities with 63.07%% of total IBD patients indicating that they take part in exercise. This is a notable observation, as previous randomized trials and correlational analyses have shown that exercise is associated with improvements in health-related quality of life in IBD patients [19, 20]. The exercise engagement levels observed in the current study are noticeably greater than those observed in earlier (i.e. prior to 2011) investigations exploring physical activity [34–36] before the body of publications concerning intervention research highlighting the efficacy of exercise in IBD [29]. However, the exercise engagement observations from this investigation are in agreement with more recent (i.e. post 2014) examinations both physical activity and exercise specific behaviours [1, 29, 37], indicating positively that the findings from randomized intervention trials and patient education information do appear to now be translating into practice.

However, (notwithstanding the distinctions between physical activity and exercise) the findings from this investigation importantly show that in relation to the general public, IBD patients appear to engage less frequently as the recent Sport England active lives survey showed that 74.2% of adults are either fairly (30–149 minutes/week) or very active (>150 minutes/week) [44]. The analyses of the barriers to physical activity indicated immoderately that a combination of factors prevent them from engaging in exercise, indicating that a multifactorial approach to improving exercise activity is necessary. Furthermore, whilst the two-way cross-

tabulation analyses showed no differences in exercise engagement between disease type, disease activity or age; female patients were significantly less likely to exercise than males, with over 40% not engaging in exercise. This is an important observation, particularly as further inspection of the cross-tabulation analyses of the question in relation to barriers to exercise did not reveal any statistical differences as a function of gender. Regardless, taking into account the positive influence of exercise on health related and indeed other quality of life indices, it is important for future implementation initiatives to be undertaken in order to enhance exercise engagement in female IBD patients.

In relation to exercise modality, this investigation indicated that moderate level exercise was the predominant aerobic modality, but the principal response in relation to resistance training was that IBD patients do not engage with this form of exercise. Furthermore, the cross-tabulation analyses showed that males were more likely to utilize machine-based resistance training whereas females were most likely to undertake structured resistance training classes. Whilst it remains unknown as to which is the most suitable resistance training modality for IBD, it is important to recognise that patients' responses also highlighted a statistical lack of knowledge in relation to the positive and negative effective effects of resistance training in IBD. These observations support the responses to the type of resistance exercise that patients engaged with. This is a prominent observation, as resistance training has been shown to enhance both bone mineral density and skeletal muscle mass in IBD, which are renowned symptoms in this patient group [31, 32]. It should be acknowledged that in relation to aerobic exercise, intervention analyses examining the effects of resistance training are far less established. Therefore, it can be speculated that limited scientific literature on resistance training in IBD, coupled with a historical emphasis on aerobic exercise, results in insufficient awareness among patients and healthcare providers, hindering engagement and knowledge dissemination regarding resistance training. It is important that further randomized intervention trials are conducted exploring the effects of resistance training on pertinent IBD indices alongside additional educational developments to translate these observations into patients own habitual disease management activities.

Further exploration of the two-way cross-tabulation analyses showed that although no differences were observed as a function of disease type, a significantly greater proportion of those with severe symptoms engage in no aerobic exercise. This observation is perhaps to be expected as intuitively it could be anticipated that those experiencing extremely pronounced disease activity would be less able/ willing to participate in exercise. Examination of the mechanisms involved in prevention from engagement in exercise appear to provide some clarification for this finding, as a significantly greater proportion of those with severe symptoms indicated that fear of increased toilet urgency was responsible for their lack of exercise engagement. This is important to note, when clinicians or exercise professionals are recommending exercise to adults with IBD. They should understand whether toilet urgency is a barrier for the patient. If toilet urgency is an issue, then recommending exercise with a known toilet location such as home-based exercise or some form of indoor exercise such as the gym could be preferential where the patient knows how long it will take to get to the toilet when starting the exercise.

As with any study, this investigation is not without limitations, which should be acknowledged. Whilst part of the aims of this investigation were to examine patients' perceptions, the findings in relation to behaviours and other important parameters such as disease activity could have been affected due to the self-reported nature of data. As IBD patients in this investigation were recruited exclusively through UK based social media advertisements, this means that like all online surveys, it is not possible to eliminate selection bias, and the results may not be directly generalizable to a wider IBD population. Of particular note, is that 76.67%% of

survey patients were female, meaning that although exercise parameters as a function of gender were considered as part of the two-way chi-squared analyses, males and in particular non-binary patients were underrepresented in this investigation.

## Conclusion

In conclusion, the current investigation adds to the current literature by exploring IBD patients' exercise perceptions, behaviours and barriers in different patient subgroups and other potentially important disease/ demographic factors. This study notably showed that significantly more participants engaged with exercise than did not, although not at the level undertaken by the general public in the UK. Examination of the barriers associated with exercise engagement revealed that a combination of factors prevents engagement in exercise, indicating that a multifactorial approach to improving exercise activity is necessary. Importantly, this study showed that females were significantly less likely to exercise than males, suggesting that future implementation initiatives be undertaken to enhance exercise participation in female IBD patients. In addition, patients with severe symptoms predominantly engaged in no aerobic exercise and denoted that fear of increased toilet urgency was responsible for their lack of exercise engagement. Therefore, it appears that to allow the benefits of exercise to be manifested, finding exercise modalities which have easy access to a toilet is required to help alleviate the stress of toilet urgency. In relation to exercise modality, this investigation indicated that moderate level exercise was the predominant aerobic modality, but the principal response in relation to resistance training was that IBD patients do not engage with this form of exercise, owing potentially to a lack of under-standing regarding the positive/ negative effects of this training modality. Given the previously observed benefits of resistance training in IBD, additional educational developments are necessary to translate this activity into patients exercise based disease management activities.

## Supporting information

**S1 Appendix.**
(DOCX)

**S2 Appendix.**
(DOCX)

**S3 Appendix.**
(DOCX)

## Acknowledgments

We would like to thank Crohn's and Colitis UK for distributing the survey.

## Author Contributions

**Conceptualization:** Jonathan Sinclair, Johanne Brooks-Warburton, Lindsay Bottoms.

**Data curation:** Jonathan Sinclair, Lindsay Bottoms.

**Formal analysis:** Jonathan Sinclair.

**Investigation:** Jonathan Sinclair, Johanne Brooks-Warburton, Lindsay Bottoms.

**Methodology:** Jonathan Sinclair, Johanne Brooks-Warburton, Lindsay Bottoms.

**Project administration:** Lindsay Bottoms.

**Writing – original draft:** Jonathan Sinclair, Lindsay Bottoms.

**Writing – review & editing:** Jonathan Sinclair, Johanne Brooks-Warburton, Lindsay Bottoms.

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
