## [Decision Letter · Decision Letter 0]

12 Dec 2023

PONE-D-23-32900Perceptions, behaviours and barriers towards exercise practices in inflammatory bowel diseasePLOS ONE

Dear Dr. Bottoms,

Thank you for submitting your manuscript to PLOS ONE. After careful consideration, we feel that it has merit but does not fully meet PLOS ONE’s publication criteria as it currently stands. Therefore, we invite you to submit a revised version of the manuscript that addresses the points raised during the review process.

While the general tone of the reviews is positive, some concerns have been raised. I would invite the authors to submit a major revision to address them. Some of the points to keep in mind are: 

1. Shorten and edit the introduction to reduce redundancy and enhance clarity;

2. Include additional information about the studied population;

3. Emphasize the novelty of the study in the discussion section;

4. Elaborate on the concept of "combination factors" and specify which factors mainly contributed to the avoidance of exercise.

We look forward to receiving your revised manuscript.

Kind regards,

Satyaki Roy, Ph.D.

Academic Editor

PLOS ONE

Journal Requirements:

2. Please ensure that you have specified a) Did participants provide their written or verbal informed consent to participate in this study?

Additional Editor Comments:

While the general tone of the reviews is positive, some concerns have been raised. I would invite the authors to submit a major revision to address them. Some of the points to keep in mind are as follows:

1. Shorten and edit the introduction to reduce redundancy and enhance clarity;

2. Include additional information about the studied population;

3. Emphasize the novelty of the study in the discussion section;

4. Elaborate on the concept of "combination factors" and specify which factors mainly contributed to the avoidance of exercise.

Reviewers' comments:

Reviewer's Responses to Questions

**Comments to the Author**

1. Is the manuscript technically sound, and do the data support the conclusions?

Reviewer #1: Yes

Reviewer #2: Partly

Reviewer #3: Partly

Reviewer #4: Yes

2. Has the statistical analysis been performed appropriately and rigorously? 

Reviewer #1: Yes

Reviewer #2: Yes

Reviewer #3: Yes

Reviewer #4: Yes

3. Have the authors made all data underlying the findings in their manuscript fully available?

Reviewer #1: Yes

Reviewer #2: Yes

Reviewer #3: Yes

Reviewer #4: Yes

4. Is the manuscript presented in an intelligible fashion and written in standard English?

Reviewer #1: Yes

Reviewer #2: Yes

Reviewer #3: Yes

Reviewer #4: Yes

5. Review Comments to the Author

Reviewer #1: The paper is about a survey, conducted among Inflammatory Bowel Disease patients in UK, concerning perceptions, behavior and barriers towards physical exercise. The argument is interesting and the study population is consistent, although with unusual great prevalence of female patients. The study results are clearly described and discussed.

However, more information could be useful concerning the so called “disease burden” as it could affect the feasibility of physical exercise besides disease activity. Ideally, the description of the studied population should include the Montreal classification (including age of disease onset, disease location and, for Crohn’s disease, the disease behavior). I understand that given the recruitment method adopted for the research this information could not be easy to collect. At least the information about previous IBD surgeries (yes/no, and if yes how many time, is a stoma present?) could be usefully added. Other easy to collect information could be the BMI: alterations of nutritional status could also have influence on physical activity.

A further minor point: line 67, tofacitinib is not correctly classifiable as a biologic drug, best defined as “small molecule”.

Reviewer #2: There are some revisions inside the manuscript like corrections, reduce the introduction and others must do it.

Reviewer #3: I read with interest the paper by Sinclair et al entitled "Perceptions, behaviours and barriers towards exercise practices in inflammatory bowel disease". It is an observational survey based on a self-admnistered, online questionnaire specifically desisgned for the purpose of the study and the enrollment was fully on social media.

In my opinion the paper is well written although different limitations are present (introduction is too long and editable, baseline characteristics of the included population are scant - for example concomitant extraintetsinal manifestations, other comorbidites ecc; I would have preferred a more detailed discussion recalling availble literature)

My fear is the lack of novelty associated to this paper where a plenty of literature is present (and in teh discussion the real novelty is not highlighted)

Reviewer #4: 1. The introduction should be shorter because of its redundancy.

2. The readers need help understanding the meaning of another IBD condition. Please give more detailed information.

3. What exactly are combination factors? Please describe in detail. Additionally, what combination of factors mainly contributed to the avoidance of exercise?

4. What are the sources of information on patients who think aerobic exercise is beneficial for IBD disease control?

5. Are there differences in response to the questions listed in Table 3 by age, between patients not on drug therapy and those who are, and between patients receiving biological therapy and those not receiving it?

6. Most IBD patients seem to know little about the significance of resistance training. Why is this? Are these perceptions different by age?

6. PLOS authors have the option to publish the peer review history of their article (what does this mean?). If published, this will include your full peer review and any attached files.

Reviewer #1: No

Reviewer #2: **Yes: **Prof. Dr. Neihaya Heikmat Zaki

Reviewer #3: No

Reviewer #4: No

---

## [Author Response · Author response to Decision Letter 0]

13 Jan 2024

Academic Editor and Reviewer Comments with Responses

Academic Editor:

Comment: Shorten and edit the introduction to reduce redundancy and enhance clarity

RESPONSE: Thank you for the suggestion, we have now shortened the introduction.

Comment: Include additional information about the studied population

RESPONSE: We have provided all of the information that we have available to use already, unfortunately this is not possible as we cannot retrospectively obtain this information as our current data is anonymized. 

Comment: Emphasize the novelty of the study in the discussion section

RESPONSE: We have now emphasized the novelty of the study within this section.

Comment: Elaborate on the concept of "combination factors" and specify which factors mainly contributed to the avoidance of exercise. Other then clarifying what the, there is no further specification that we can give that it not already elucidated within the paper already based on the data that we have.

RESPONSE: This has now been added.

Reviewer 1:

Comment: More information could be useful concerning the so called “disease burden” as it could affect the feasibility of physical exercise besides disease activity. Ideally, the description of the studied population should include the Montreal classification (including age of disease onset, disease location and, for Crohn’s disease, the disease behavior). I understand that given the recruitment method adopted for the research this information could not be easy to collect. At least the information about previous IBD surgeries (yes/no, and if yes how many time, is a stoma present?) could be usefully added. Other easy to collect information could be the BMI: alterations of nutritional status could also have influence on physical activity.

RESPONSE: Unfortunately, although this could potentially be useful information to the paper, as the data/ questionnaire responses are anonymized, it is impossible to retrospectively collect this additional information.

Comment: A further minor point: line 67, tofacitinib is not correctly classifiable as a biologic drug, best defined as “small molecule”.

Response: You are correct, we have now amended this. 

Reviewer 2:

Comment: There are some revisions inside the manuscript like corrections, reduce the introduction and others must do it.

RESPONSE: We have now reduced the volume of the introduction section.

Reviewer 3:

Comment: In my opinion the paper is well written although different limitations are present (introduction is too long and editable, baseline characteristics of the included population are scant - for example concomitant extraintetsinal manifestations, other comorbidites etc; I would have preferred a more detailed discussion recalling available literature)

RESPONSE: We have now reduced the volume of the introduction section. 

Unfortunately, with regards to the baseline characteristics, although this could potentially be useful information to the paper; as the data/ questionnaire responses are anonymized, it is impossible to retrospectively collect this additional information.

With great respect, we do not agree in relation to the discussion section.

My fear is the lack of novelty associated to this paper where a plenty of literature is present (and in teh discussion the real novelty is not highlighted)

RESPONSE: We have now highlighted the novelty of the paper to a greater extent within the discussion section at yours and other reviewers’ recommendation. 

Reviewer 4:

Comment: The introduction should be shorter because of its redundancy.

RESPONSE: We have now reduced the volume of the introduction.

Comment: The readers need help understanding the meaning of another IBD condition. Please give more detailed information.

RESPONSE: Although Ulcerative Colitis and Crohn’s represent the overwhelming majority of cases of inflammatory bowel disease, there are other inflammatory bowel conditions such as Microscopic colitis and it is also possible for patients who suffer from both Crohn’s and Ulcerative Colitis. Therefore, the ‘other’ category was present to represent such patients. Given the 11 respondents selected this option, we feel that doing so was a valid decision. This has now been communicated within the paper. 

Comment: What exactly are combination factors? Please describe in detail. Additionally, what combination of factors mainly contributed to the avoidance of exercise?

RESPONSE: The ‘combination’ of factors was created as patients were able to select multiple items on some questions, therefore, to be able to undertake the necessary chi-squared analyses which require categorical data, those who did so were categorized into a ‘combination’ category. The questions where respondents could select multiple options was already shown in Supplementary materials 3 but how the ‘combination’ variables were calculated has now been articulated within the paper. 

Comment: What are the sources of information on patients who think aerobic exercise is beneficial for IBD disease control?

RESPONSE: Participants will have had access to general information about physical activity as does the public. There will be nothing specific for those with IBD. 

Comment: Are there differences in response to the questions listed in Table 3 by age, between patients not on drug therapy and those who are, and between patients receiving biological therapy and those not receiving it?

RESPONSE: There were no differences as a function of age or drug therapy as was already outlined in the previous submission under the ‘Exercise perceptions, beliefs and behaviours (two-way cross-tabulation analyses’ heading. 

As stated already within statistical analyses section of the paper we tested differences in responses to each question based on baseline/demographic variables, i.e. gender, age, specific IBD condition and disease activity etc, yet only statistical analyses that exhibited statistical significance are presented within the results. This was undertaken in order to improve the clarity of the results and give precedence to the statistically significant observations – the results section would be around 4-5 times longer if we wrote about all of the non-significant findings. 

Comment: Most IBD patients seem to know little about the significance of resistance training. Why is this? Are these perceptions different by age?

RESPONSE: Unfortunately, as we did not specifically examine ‘why’ IBD patients know little about the importance of resistance training we only examined the questions in relation to the type of resistance training, whether respondents thought that resistance training was positive/ negative and avoidance of certain types of exercise, it is not possible to determine why this was the case. We have however, attempted to the greatest extent possible without making unsubstantiated claims why the aforementioned observation might be the case. 

As per above in relation to age, the responses did not differ as a function of age or indeed any other of the baseline/ demographic parameters, in addition to what is already stated within the paper.

---

## [Decision Letter · Decision Letter 1]

6 Feb 2024

Perceptions, behaviours and barriers towards exercise practices in inflammatory bowel disease

PONE-D-23-32900R1

Dear Dr. Bottoms,

We’re pleased to inform you that your manuscript has been judged scientifically suitable for publication and will be formally accepted for publication once it meets all outstanding technical requirements.

Kind regards,

Satyaki Roy, Ph.D.

Academic Editor

PLOS ONE

Additional Editor Comments (optional):

The authors have addressed all concerns. I recommend accepting the manuscript.

Reviewers' comments:

Reviewer's Responses to Questions

**Comments to the Author**

1. If the authors have adequately addressed your comments raised in a previous round of review and you feel that this manuscript is now acceptable for publication, you may indicate that here to bypass the “Comments to the Author” section, enter your conflict of interest statement in the “Confidential to Editor” section, and submit your "Accept" recommendation.

Reviewer #1: All comments have been addressed

Reviewer #3: All comments have been addressed

Reviewer #4: All comments have been addressed

2. Is the manuscript technically sound, and do the data support the conclusions?

Reviewer #1: Yes

Reviewer #3: Yes

Reviewer #4: Yes

3. Has the statistical analysis been performed appropriately and rigorously? 

Reviewer #1: Yes

Reviewer #3: Yes

Reviewer #4: Yes

4. Have the authors made all data underlying the findings in their manuscript fully available?

Reviewer #1: Yes

Reviewer #3: Yes

Reviewer #4: Yes

5. Is the manuscript presented in an intelligible fashion and written in standard English?

Reviewer #1: Yes

Reviewer #3: Yes

Reviewer #4: (No Response)

6. Review Comments to the Author

Reviewer #1: The Authors’ answers to the reviewers’ comments are sufficient. The modifications to the manuscript make it suitable for publication.

Reviewer #3: Dear Authors,

thank you for your reply and the effort.

I think you have fully answered to all the comments

Reviewer #4: The authors appropriately responded to this reviewer's comments.

Unfortunately, the authors should accumulate more information of IBD patients in this study.

7. PLOS authors have the option to publish the peer review history of their article (what does this mean?). If published, this will include your full peer review and any attached files.

Reviewer #1: No

Reviewer #3: **Yes: **Stefano Festa

Reviewer #4: No

---

## [Editor Report · Acceptance letter]

22 Mar 2024

PONE-D-23-32900R1 

PLOS ONE

Dear Dr. Bottoms, 

I'm pleased to inform you that your manuscript has been deemed suitable for publication in PLOS ONE. Congratulations! Your manuscript is now being handed over to our production team.

Kind regards, 

on behalf of

Dr. Satyaki Roy 

Academic Editor

PLOS ONE